# 3D-Printed Multilayer Sensor Structure for Electrical Capacitance Tomography

**DOI:** 10.3390/s19153416

**Published:** 2019-08-04

**Authors:** Aleksandra Kowalska, Robert Banasiak, Andrzej Romanowski, Dominik Sankowski

**Affiliations:** Institute of Applied Computer Science, Lodz University of Technology, 90-924 Lodz, Poland

**Keywords:** 3D, ECT, 3D-printing, sensors, modeling

## Abstract

Presently, Electrical Capacitance Tomography (ECT) is positioned as a relatively mature and inexpensive tool for the diagnosis of non-conductive industrial processes. For most industrial applications, a hand-made approach for an ECT sensor and its 3D extended structure fabrication is used. Moreover, a hand-made procedure is often inaccurate, complicated, and time-consuming. Another drawback is that a hand-made ECT sensor’s geometrical parameters, mounting base profile thickness, and electrode array shape usually depends on the structure of industrial test objects, tanks, and containers available on the market. Most of the traditionally fabricated capacitance tomography sensors offer external measurements only with electrodes localized outside of the test object. Although internal measurement is possible, it is often difficult to implement. This leads to limited in-depth scanning abilities and poor sensitivity distribution of traditionally fabricated ECT sensors. In this work we propose, demonstrate, and validate experimentally a new 3D ECT sensor fabrication process. The proposed solution uses a computational workflow that incorporates both 3D computer modeling and 3D-printing techniques. Such a 3D-printed structure can be of any shape, and the electrode layout can be easily fitted to a broad range of industrial applications. A developed solution offers an internal measurement due to negligible thickness of sensor mount base profile. This paper analyses and compares measurement capabilities of a traditionally fabricated 3D ECT sensor with novel 3D-printed design. The authors compared two types of the 3D ECT sensors using experimental capacitance measurements for a set of low-contrast and high-contrast permittivity distribution phantoms. The comparison demonstrates advantages and benefits of using the new 3D-printed spatial capacitance sensor regarding the significant fabrication time reduction as well as the improvement of overall measurement accuracy and stability.

## 1. Introduction

Electrical Capacitance Tomography (ECT) is a non-intrusive and non-invasive imaging modality dedicated to monitoring industrial processes in pipelines and reactors, wherever non-conducting dielectric materials’ mixtures are processed. Typical examples may be shown as follows: gas-oil flows [1], solid particle flows [2] or reservoirs [1,3,4,5]. A typical ECT system measures mutual capacitance changes between pairs of electrodes distributed around the circumference of an industrial process container or pipe [6,7,8,9]. The amount of measurement data depends on used hardware (typically 66–496) and its acquisition rate may vary from a few to hundreds of images per second [6,9]. Thereafter, collected data can be processed by a high-performance PC system using mathematical modeling [5,10] and specialized algorithms for reconstruction of images [5,11,12,13], analyzing raw data [3,8,14,15] and finally making a right diagnostic decision for process control and automation [16,17,18].

Industrial processes generally have a 3-dimensional nature. Hence, it is an obvious tendency to sense and measure phenomena in real 3D space occupied by these industrial processes. Therefore ECT tomography is evolving from the *z*-axis averaged measurement recorded in the cross-sectional plane to the 3D scanning of the entire volume [8,13,16,19,20,21,22]. Classical ECT measurement systems employ a single-layer regular electrode layout for cross-sectional sensing. Image reconstruction processes strongly rely on a quality of the measurement sensitivity analysis [5,16,23,24,25]. Sensitivity analysis focuses on the modeling of an electric field inside the sensing space volume. Hence, it tends to minimize the reconstruction approximation error for the given measurement data set. However, 3D ECT sensors concepts extend the notion of the reconstruction from the plane cross-sectional imaging to the fully volumetric reconstruction of the given material distribution in the entire sensor space as has been reported in [2,23,24,26,27]. This work reveals the new concept of various ECT sensor 3D structure arrangements. The main components of 3D ECT systems are illustrated in Figure 1.

This research work is devoted to a new type of 3D ECT sensor structure where its mounting support is completely fabricated using 3D printing technology. A comparison of 3D-printed ECT sensor and traditionally fabricated sensor will be performed in to investigate their detectability for low- and high-contrast dielectric phantoms. The comparison demonstrates differences and similarities of using the new 3D-printed spatial capacitance sensor over a traditionally fabricated device.

## 2. Theoretical Fundamentals of 3D ECT Sensor Modeling

3D capacitance tomography sensor modeling is typically named as a forward problem. Forward problem-solving techniques employ simulation of output measurement data for a given set of excitation values as well as permittivity values characterizing the dielectric material distribution. The inverse problem can be considered to be an imaging result or a raw analysis for a given dataset of measurement records. In 3D ECT tomography data workflow, the forward problem must be solved before the inverse problem solution.

To solve a 3D ECT forward problem, two assumptions can be made. The first one is no wave effect. The second one is a low-frequency approximation approach to Maxwell’s equation-solving. Therefore in a simplified mathematical 3D ECT model, the following electrostatic approximation (Equation (Equation 1)):(1)0=∇×E
is made. This approach leads to neglecting the effect of wave propagation. Taking (Equation (Equation 2)):(2)E=−∇φ
one can assume no internal charges. Hence, the following Equation (Equation 3) holds:(3)∇ε(x,y,z)∇φ=0
where: φ is a spatial distribution of electric potential inducted by electric field *E*, ε(x,y,z) is a spatial dielectric permittivity distribution. The potential φ of each electrode can be derived from Equation (Equation 4):(4)$$\varphi =\nu_{k}\;at\;elec_{k}$$
where: $elec_{k}$ is the *k*th electrode at the potential $\nu_{k}$. To solve this equation, one can apply a variety of numerical methods. In this work, the finite element method (FEM) method is employed to derive (Equation (Equation 5)):(5)Y(ε)φ=F
where: the matrix *Y* is the discrete representation of the operator ∇ϵ∇ including the Dirichlet-type boundary conditions, the vector *F* is the Neumann type boundary condition term and φ is the approximated electric potential solution. The total electric charge *Q* on the electrode $e_{k}$ is given by Equation (Equation 6):(6)$$Q_{k}=\int_{e_{k}}\varepsilon \frac{\partial \varphi }{\partial n}dx^{2}$$
where: *n* is the inward normal on the *k*th electrode.

The FEM is recognized as an efficient computational tool that can be applied for a 3D ECT sensor design and modeling process to simulate an interior electric field dispersion [24]. It may significantly help for the capacitance value estimation as well as sensitivity maps to optimize sensor geometry and structure. The sensitivity maps, dependent on the sensor geometry, are essential characteristics of the 3D ECT systems [2]. One of the ECT sensing systems features an extremely small sensing ability in the middle of the sensor, due to the incapitalization capacitance values can be very small for most distant layers, typically in the range of 0.01–1 pF [13] and it differences with low and high-contrast media the dimensions of the electrodes must be a trade-off between their heights and mutual capacitances. It is necessary to achieve a measurement range comparable with measurement range of the used system hardware. Homogeneous sensitivity distribution inside a sensing domain is also an electrical field intensity uniformly inside an investigated volume [2,13,21,28].

The common 3D ECT sensor is an array of metal-plate electrodes arranged around the sensed medium, typically on an outer boundary of a non-conducting pipe. External, grounded potential shield keeps the electric field lines inside the sensor space. In some cases of conducting reactors, such as metal pipes or containers, the electrodes are located on the inside surface of the reactor. Whenever this is the case, the metal walls are then regarded as the electric shielded layer with the other external coupling components such as the radial and guard electrodes, to improve the quality of measurements and hence the reconstruction process.

## 3. 3D Modeling & Printing of ECT Capacitance Sensors

3D printing usually refers to the additive type of manufacturing 3D objects. This process is conducted with use of 3D printers following detailed, computer-aided scripts generated from a digital model prepared for fabrication processes. The resulting object is produced in an iterative process of adding consecutive layers of melted substance on top of another. This work presents a design and outcomes of a spatial capacitance sensor fabricated with the aid of the Fused Filament Fabrication technique. To develop a new mechanical structure of precise 3D ECT sensor, its computational design must be prepared first. The Blender modeling software suite was employed to prepare a complete design of a 3D sensor structure. We used the stereolithography (STL) format output from the Blender software. The mechanical design model for 3D printing was developed under several important constraints, especially in accordance to the mesh structure design guidelines. The internal support mesh structure was strictly validated for thin-wall issues, redundancy required for stiffness, layered production compliance, etc.

The proposed design of the sensor was of substantial size exceeding standard equipment maximum print height in vertical direction. Hence, it was decided to divide the sensor print into 2 distinct elements. The maximum print height for Ultimaker 3, employed in this study, is 200 mm. The final design was rearranged into 2 symmetrical parts, therefore some additional joint elements (insets and holes were added at the contact edges) were required to connect both elements after the printing. Additionally, walls separating electrode layouts at the outside external side were printed at half thickness to ensure tight fit and uniformity of the 3D sensing structure.

The next step was to prepare the 3D printing configuration. In this study, the Ultimaker 3 printer with PLA (polylactic acid) filament and Ultimaker CURA 4.1 printing software were used for the building and controlling of the printing process. It is possible to construct detailed spatial ECT sensor arrangements of variable shapes and in a wide range of dimensions using 3D printing equipment. The maximum resolution of the used 3D printer is 0.4 mm for xy-axes and 0.06 mm for *z*-axis (print layer thickness) and can be extended by using extended-resolution extruders. A high XY resolution was used to generate a very thin sensor wall for keeping the distance between electrodes and scanned volume as low as possible. In the past it was very hard or even impossible to achieve this in previous traditionally manufactured PMMA (or PVC)-based 3D ECT sensors. In this research, the PLA filament has been used as a widely used popular 3D printing solution with its relative permittivity value 3–5 at 1 MHz [29]. The PLA filament provides a good balance between durability, flexibility, and ease to use. Its dielectric properties are well-fitted to the dielectric properties of tested phantoms. For 3D ECT printed sensor, a PLA extruding temperature is worth raising by 5–10 ${}^{\circ }$C to provide better Z-direction bonding. The one disadvantage is that PLA is characterized by a relatively low temperature, in which it is losing its stiffness and durability (60 ${}^{\circ }$C). This must be taken into consideration when 3D ECT sensor industrial applications are developed. As a much stronger alternative, a nylon filament can be used as a printing material, but its relative permittivity value is slightly lower than PLA–2.5 at 1 MHz. A nylon 3D print offers even better durability and it is more resistant to higher temperatures (>150 ${}^{\circ }$C). However, ABS-based filaments are too breakable to be used for 3D ECT sensor manufacturing, however their temperature resistance is relatively high (>180 ${}^{\circ }$C). True three-dimensional ECT sensor arrangement needs a specific design with additional walls that separate electrodes working in an inter-measurement plane mode. As a result, a dual extruder mode had to be used to build extra support. PVA water-dissolvable material was used to produce these supplementary supports. PVA is a unique filament that works properly with PLA, nylon, and other filaments with similar melting temperature. As was mentioned earlier, main electrode placements have been optimized to be as thin as possible (0.2 mm) to strengthen electric field penetration abilities by using a single layer of filament. However, more layers (2–3) can be applied as well to make a sensor pipe stiffer and more durable. The increased number of filament layers with fixed thickness (1 mm) has been used to build inter-plane separators for horizontal and vertical screening systems. Eletrical components were added to the finished prefabricated structure in the form of wiring, electrodes, and shielding. Figure 2 presents the workflow diagram for 3D printing of ECT sensor structures.

## 4. Experimental Setup

Experiments and measurement was conducted in Tom Dyakowski Process Tomography Lab at Lodz University of Technology, Poland. Verification of the classical vs. proposed 3D printed ECT sensor was based on two distinct testing concepts. The first strategy was to use the precise Agilent E4980A impedance meter combined with 64-channel computer-controlled multiplexer for low-contrast solid-gas mixtures (objects) in offline mode. The LCR meter-based system can precisely detect signals that can be useful in capturing extremely small capacitance value changes in spatial measurements for distant ECT sensor electrodes. In this paper, an LCR meter-based experimental system was directly used for investigating low-permittivity contrast mixtures that can be challenging objects in terms of accurate measurement of spatial low-amplitude (<1 pF) capacitance value changes. For this approach, a single 496-measurement raw data frame was acquired three times: Step 1—for low-permittivity distribution (empty sensor); Step 2—for high permittivity (a sensor was completely fulfilled by PE granules) and Step 3—for the investigated object. The second approach was to use a specialized charge–discharge principle-based 32-channel ECT measurement system (ET3) with flexible two-way gain amplifier setup. ET3 measurement hardware was applied in online mode for high-contrast dielectric liquid–gas mixtures. This strategy used the internal ET3 calibration procedure and generated normalized capacitance data. For 32-channel, the ET3 system was able to acquire 11 frames per second. As with the LCR-based approach, three steps of data acquisition were performed as well. By contrast, 50 frames at each step were taken and averaged to smooth a signal and reduce a measurement noise. Both measurement systems used in this study are presented in Figure 3.

The 3D capacitance-measurement sensors used in this test was arranged of 32 electrodes grouped in four cylindrical layers consisting of 8 electrodes in each one [12]—Figure 4.

Each of those sensors included two border planes (1st and 4th) and two inner planes (2nd and 3rd). Therefore, they have met major requirements of the spatial capacitance-measurement principle which need typically a significant number of 16–32 electrodes to capture an industrial process with reasonable axial and spatial resolution. To obtain performance of studied ECT sensors, the intra-layer and inter-layer electrode excitation strategy was used to provide *m* = 496 independent capacitance data for $n_{el}=32$ excluding repeating mutual measurements (Equation (Equation 7)):(7)$$\begin{array}{c} m=\frac{n_{el}\left(n_{el}-1\right)}{2} \end{array}$$

The traditionally made 3D ECT sensor (we name it here as S1) has already been developed and verified in the past [10,19]. Thus, in this study we positioned it as a baseline reference capacitance sensor for all tests. S1 was designed and built with the aid of the classical, “hand-made” method. S1 is built out of PMMA (polymethyl methacrylate) off-the-shelf pipe. S1 has the following dimensions: external diameter of 158 mm, the total height of 304 mm and pipe thickness of 4 mm with slight variations over the length (±2 mm) due to the inaccurate PMMA fabrication process. The sensor was asymmetrical in terms of the electrode layers arrangement: outer electrode height equals 70 mm and inner electrode only 30 mm. The electrode layers were placed on the outer surface of the sensor which led to external type of measurement. Sensor shielding was composed of double 25 mm guard screens located on the top and bottom parts of the structure plus extra, outer protective screens for external electric interference.

A novel 3D printed sensor (we name it here as S2) had same dimensions as S1 and was developed using a novel 3D modeler design and 3D-printing technology. PLA (transparent material) was employed for building the internal mechanical support sensor structure. The mounting pipe thickness of S2 was only 0.4 mm and constant over the pipe wall. It simultaneously provided the insulation between the electrode’s active surface and industrial process and at the same time preserved deep measurement penetration. Additionally, this arrangement can enhance sensitivity in the central area of a sensor. The new concept of S1 included a copper internal radial screen between adjacent electrodes in the same layer and between adjacent layers. This prevented the electric field uncontrolled leakage for excited electrodes and kept the electric field distribution more uniform. This idea was an extension to some previous research using guard electrodes in ECT [28]. Using a new 3D-printing approach, it was easy to integrate such an advanced screening structure within electrodes—see Figure 5.

To investigate the low-contrast medium, a set of three cylindrical 3D printed phantoms (objects) was developed (TestA, TestB, and TestC)—Figure 6. The test phantoms were constructed using the 3D-printing technique (PLA) and filled with 3 mm PE (polyethylene) granules in two ways—see Table 1.

For capacitance characteristics and standard deviation evaluation, raw capacitance-measurement data taken from the LCR meter were normalized using high $C_{full}$ and low $C_{empty}$ permittivity distributions (PE granulate dielectric constant ≈3.2 at room temperature, with the air dielectric constant = 1—as the background).

In this study, the normalization of experimental capacitance data $C_{n}$ was computed according to formula [4]:(8)$$\begin{array}{c} C_{n}=\frac{C-C_{empty}}{C_{full}-C_{empty}} \end{array}$$

To evaluate performance of both sensors for high-contrast objects (gas–liquid mixtures) a set of ten cylindrical-shaped 3D printed phantoms were developed—Figure 7. The test phantoms were constructed using PLA filament and printed using the “spiral vase” option to avoid liquid leakages. As a test liquid, the propylene glycol (dielectric constant ≈32 in room temperature) was used, while the air acted as a background. All the high-contrast phantoms were designed as cylinders with various diameters (10–40 mm). The shapes, diameters, and positions of these test objects were somehow similar to the previous low-contrast set, but its roles were different. In this study, only the high-contrast material “liquid inside a cylinder object” option was considered to investigate small (10 mm, 15 mm, 20 mm) and middle-size (2 × 40 mm) high-contrast cylinders. Small-diameter cylinders were located in three positions (70 mm—geometrical sensor center, 35 mm from geometrical sensor center, 60 mm from geometrical sensor center) on the path along the 3D ECT sensor radius. The middle-size diameter object included two 40 mm cylinders and it was located symmetrically in two positions along the sensor diameter at equal distance from the sensor center (35 mm). The arrangement of the specific electrode area and height asymmetry relation for outer and inner layers was preserved for both sensors to retain the distant measurement signal for the inter-plane electrodes (for instance, the electrode pair: 1–29) at a level possible to be sensed by the ECT measurement unit. The following measurement workflow was applied. 15 V positive potential was excited on the sender electrode with the other electrodes grounded. Then consecutively each electrode was switched to be the sender, one by one.

## 5. Results and Discussion

### 5.1. Low-Contrast Objects Investigation

In this section, the comparison of detection capability for sensors S1 and S2 is discussed in six experimental conditions of low-contrast (low permittivity distribution) solid-gas phantom objects. To analyze and compare the measurement abilities of the tested sensors, four capacitance-measurement inner-layer and inter-layer configurations for acquiring RCD (Raw Capacitance Data) have been selected: 1–2 (Layer1–Layer2), 1–9 (Layer1–Layer2), 1–17 (Layer1–Layer3), and 1–25 (Layer1–Layer4) for 6 various tested low-contrast objects—see Table 2.

A capacitance measurement has been conducted using precise LCR impedance analyzer and multiplexer hardware and a set of 6 capacitance characteristics for full 496 measurements cycle has been captured and normalized using low (for the air) and high (for PE granules) reference measurements. A standard deviation value of all normalized capacitance characteristics was calculated as a “signal stability” indicator for evaluating the performance of the S1 and S2 sensor.

For most tests sensor S2 with configuration S2-1-17 achieved similar capacitance-measurement value to S2 for both close-to-wall and central area. For most distant configurations 1–25 an advantage S2 over S1 could be observed. S1-1-2 and S2-1-2 configurations resulted in the maximum value of capacitance measurement as it connects adjacent electrodes from layer 1—those electrodes had the biggest active surface. In this case, S1 seemingly offered much higher value of capacitance measurement than S2 for the whole set of testing phantoms. A similar tendency existed when configurations S1-1-9 and S2-1-9 were taken into consideration. It was a consequence of high electric field intensity peak value between two adjacent electrodes in S1, which were not separated by vertical radial screens. Vertical radial screens in S2 significantly reduced the electric field intensity value and kept a signal at a lower level. The same trend could be observed for configurations S1-1-9 and S2-1-9, where two adjacent electrodes from two adjacent sensor layers were connected. When we analyzed more distant electrode pairs, a real advantage of sensor S2 over S1 could now be observed.

Figure 8, Figure 9, Figure 10, Figure 11, Figure 12 and Figure 13 presented selected 1st electrode measurement cycle extracted from the full capacitance data set for sensor S1—blue graph and S2—red graph. These normalized capacitance characteristics have been computed for all phantom Test*A*-Test*C* configurations using the 3D ECT calibration procedure. The most representative Test*B* phantom was investigated here to check the ability of both sensors to detect low-contrast permittivity distribution with an approximate equal distance from electrodes and from sensor center along axis *z*. The mean value of the normalized capacitance records dataset for S1-Test$B_{out}$ was 1.196, and the standard deviation was 0.181. The mean value of the normalized capacitance dataset for S2-Test$B_{out}$ was 1.098 and the standard deviation was 0.115. Therefore, this study proved that S2 outperformed S1.

The S2 capacitance-measurement signal fitted better than the calibration limits and was more stable than S1. The Figure 10 presented 1st electrode measurement cycle graph extracted from full capacitance data set for sensor S1—blue graph and S2—red graph, obtained for phantom Test$B_{in}$. This phantom demonstrated that both sensors are capable of detecting permittivity distribution in the center of the sensing area. For this test, the mean value of normalized capacitance dataset for S1 was 0.686 and the standard deviation was 0.396. The mean value of the normalized capacitance dataset for S2 was 0.721 and the standard deviation was 0.394. A capacitance-measurement data from S2 better fit the calibration limits and suffered from rapid changes of capacitances from specific electrode pairs. This test confirmed again an advantage of S2 over S1. The same trend could be observed during the remaining experimental tests for other low-contrast phantoms Test*A* and Test*C*.

### 5.2. High-Contrast Objects Investigation

High-contrast (high permittivity distribution) gas–liquid object tests were focused on axial and spatial resolution abilities of both investigated 3D ECT sensors. The sizes of phantom diameters were selected to observe a point of object detectability, especially in the central region of devices where an electric field power is degraded significantly. Due to the static nature of tested objects, a temporal resolution was not taken into consideration in this study. As with the previous low-contrast evaluation, a standard deviation value of normalized capacitance characteristics was considered to be a “signal stability” indicator for evaluating the performance of S1 and S2 sensor. Additionally, a mean value of normalized capacitance characteristics was computed to determine a relative level of calibrated signal response to a background permittivity distribution in the presence of high-permittivity objects. The higher mean value would lead to a better detectability of smaller objects in various 3D ECT sensor areas. Table 3 and Table 4 present the results of a standard deviation and mean values obtained for the 1st electrode measurement cycle (in turn with 2..32 electrodes) for both sensors S1 and S2 and all sets of test objects. To compare values, a percentage difference was calculated.

The standard deviation indicators (std) proved that sensor S2 is more stable (1.5–83%) and accurate for the majority of conducted tests except 2 × 40 and 10${}_{P1}$ tests where standard values were higher for S2 and the highest at all. These two situations only occurred for extreme cases: the largest high-contrast inclusion and the smallest one located in the less-sensitive region of the sensor. It may happen when the ET3 system range limits are achieved for a given setup of its channel amplifier gains. The optimization of channel gains for signal borderlines could help to stabilize the signal for both sensors and improve S2 3D ECT sensor overall performance.

While analyzing the general mean value of the acquired normalized signals for the 1st electrode measurement cycle, it was difficult to indicate the clear winner of this comparison except in very important tests 10${}_{P1}$–10${}_{P3}$. All of them demonstrated better high-contrast object detection abilities of S2 sensor. These particular objects represent the smallest high-contrast inclusion. Test 10${}_{P1}$ was only passed by sensor S2. S2 generated output signals that could even be usable for image reconstruction algorithms. For this specific test, S1 signal response was too low and too noisy to be distinguished from lower-calibration limit (background). The remaining mean values for a given senor depended on the specific test. For 20 mm objects, there was an advantage of S2 printed sensor over S1 while this situation changed for all 15 mm objects. However, the percentage dominance for 15 mm object is on the side of the S1 sensor, and the absolute mean value for 10 mm and 15 mm objects is very low and may contain some amount of measurement noise. To look precisely at how each of the 3D ECT senor signal responses looks like a set of normalized capacitance characteristics for all test objects was computed, see Figure 14, Figure 15, Figure 16, Figure 17, Figure 18, Figure 19, Figure 20, Figure 21, Figure 22 and Figure 23.

Most of these characteristics pointed out the advantage of S2 over S1. However, the signal profile for position P1 was not easy to compare due to measurement noise (Figure 15, Figure 18 and Figure 21) the better signal response of S2 was visible for object position P2—(Figure 16, Figure 19 and Figure 22) and clearly visible for object position P3 (close to the sensor internal wall) as well—(Figure 17, Figure 20 and Figure 23).

The conducted experiments showed that new 3D printed ECT sensor noticeably outperforms the traditionally fabricated sensor when low- and high-contrast small-sized objects (10–15 mm) located in the central region of tomographic inspection were under test. For large-sized high-contrast objects (20 mm and 2 × 20 mm) the performance improvement depended on its radial and axial position. However, both sensors offered similar performance for all the tested objects in the neighborhood of the electrodes where the electric field and sensitivity were strong enough to compensate wall thickness differences.

## 6. Conclusions and Directions for Further Work

In this paper, we demonstrated and validated a new approach for fabrication of ECT sensors. This work comprises the complete workflow for design, modeling and development of the ECT sensors dedicated to true 3D measurement. The experiments showed high precision and good performance of the proposed 3D ECT sensor concept. The proposed method employs modern 3D printing technique that offers efficient ways for the manufacture of capacitance sensors. The developed 3D printed ECT sensor has taken the most valuable features and advantages from traditionally hand-made 3D ECT constructions as well as features possible to gain from a 3D-printing technique. Output sensing structures have thin-wall levels between the electrode arrays and testing volume as well as advanced structure of internal vertical and horizontal screening system. The achieved high accuracy of the design and structure results from high-resolution 3D-printing technology. Using the 3D-printing principle, a majority of 3D ECT sensor structures can be easily designed and 3D-printed in one working day. In this study, the novel 3D-printed 3D ECT sensor model has been compared to the representative solution used widely in the past. For low-contrast and high-contrast objects, we achieved at least similar or better 3D-printed sensor overall performance. Conducted experiments showed a noticeable potential of 3D printing technology applied for the fabrication of capacitance sensors. The research presented in this study is a step forward towards high-precision and high-resolution volumetric ECT systems. The directions for future development derived from this work comprise testing sensors with a larger number of electrodes, and developing novel shapes of electrode arrays coupled with novel shielding systems. The interesting implications of the 3D-printing technique is the possibility to tailor specific designs suited to specific applications such as batch crystallization or gas–liquid separation. The ultimate challenge is to incorporate the proposed approach to measurement data problems that are still under intensive research with no easy automatic way of processing such as crowd-sourcing processing of imagery data as shown in [30] or for modern contextual data processing frameworks as shown in [17,31].

## Figures and Tables

**Figure 1 sensors-19-03416-f001:**
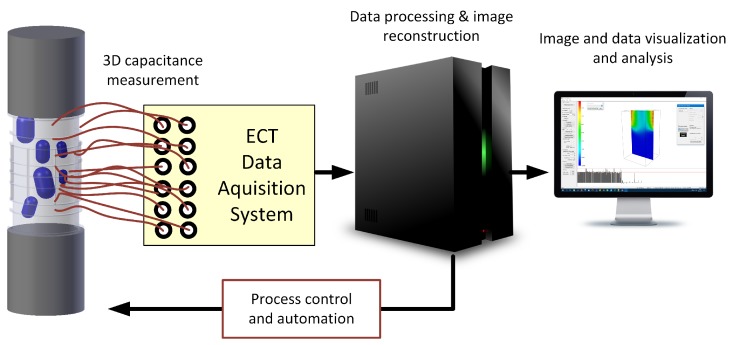
3D ECT system and its components.

**Figure 2 sensors-19-03416-f002:**
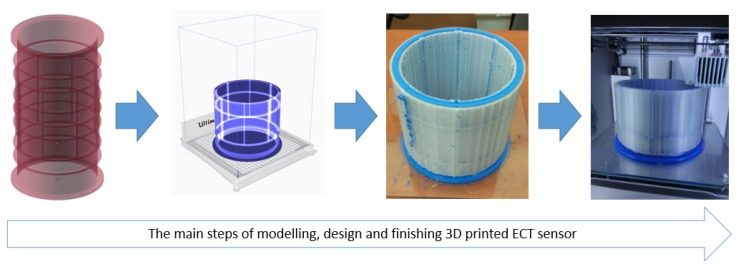
3D printed ECT sensor building workflow.

**Figure 3 sensors-19-03416-f003:**
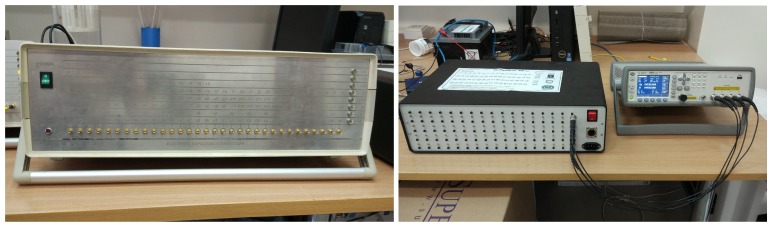
Experimental setup hardware: left—32-channel ET3 measurement hardware, right—Agilent E4980A with 64-channel multiplexer device.

**Figure 4 sensors-19-03416-f004:**
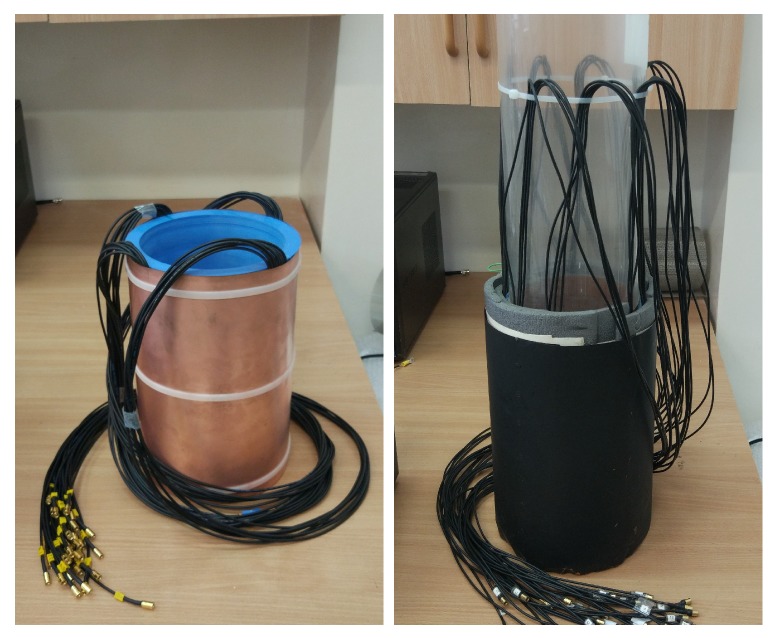
Pictures of two experimental constructions of 3D ECT sensors under study.

**Figure 5 sensors-19-03416-f005:**
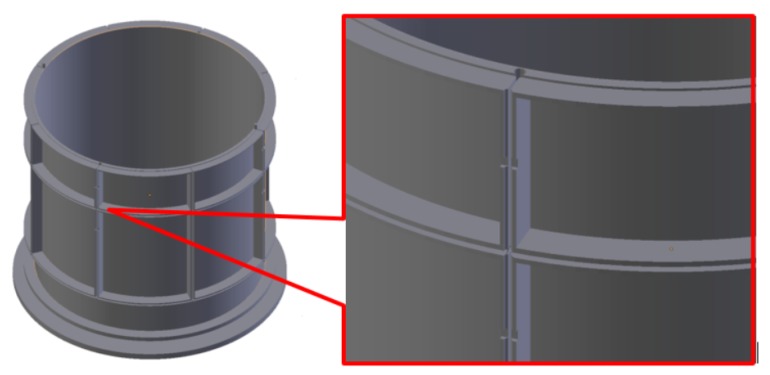
A concept of horizontal and vertical internal screening for 3D ECT sensor structure.

**Figure 6 sensors-19-03416-f006:**
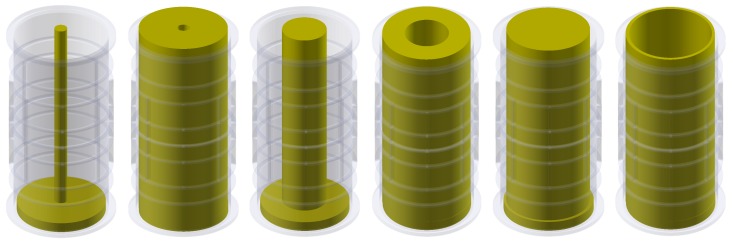
Arrangement of tested low-contrast objects according to Table 1—from the leftmost Test*A*, Test*B*, Test*C*.

**Figure 7 sensors-19-03416-f007:**
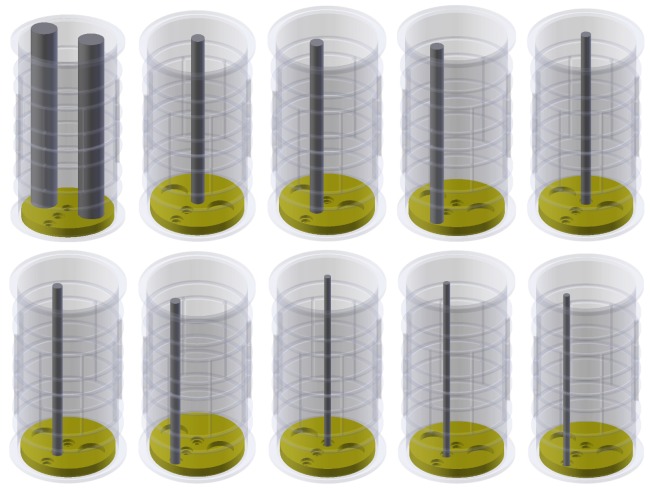
A set of ten phantoms used during high-contrast media measurements tests. The mounting stand had 5 holes. Three 10 mm of diameter holes were positioned along the sensor profile diameter at given positions: “P1” at x = 70 mm, y = 70 mm; “P2” at x = 70 mm, y = 40 mm; “P3” at x = 70 mm, y = 10 mm. Two additional 40 mm of diameter holes were positioned symmetrically at x1 = 110 mm and x2 = 40 mm for y = 70 mm. All the rods were parallel to sensor walls.

**Figure 8 sensors-19-03416-f008:**
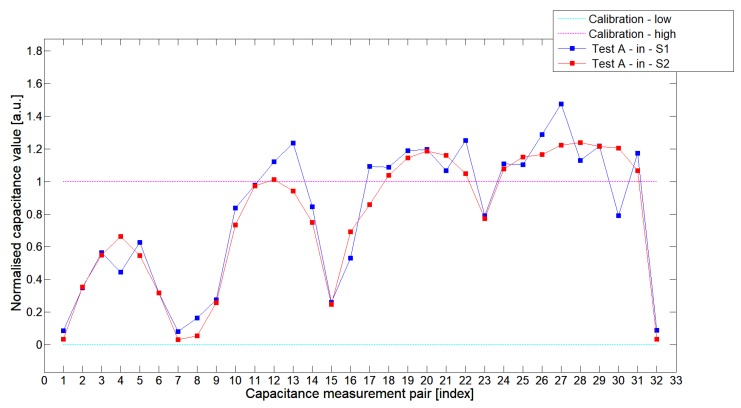
The 1st electrode measurement cycle (S1: 1->32 and S2: 1->32)for Test$A_{in}$ and S1—blue line, S2—red line. Cyan and magenta lines indicate calibration limits (0;1).

**Figure 9 sensors-19-03416-f009:**
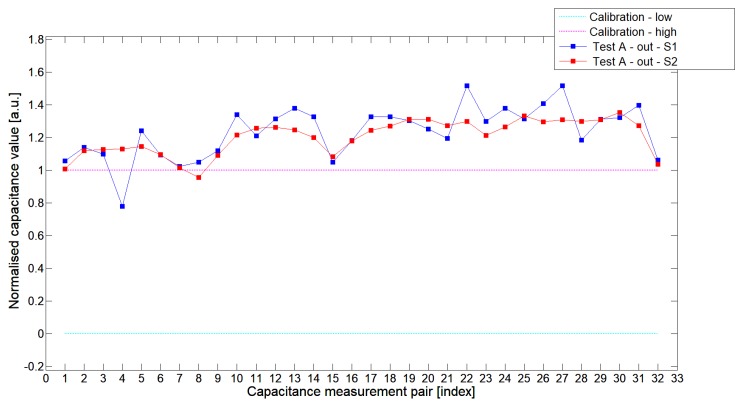
The 1st electrode measurement cycle (S1: 1->32 and S2: 1->32) for Test$A_{out}$ and S1—blue line, S2—red line. Cyan and magenta lines indicate calibration limits (0;1).

**Figure 10 sensors-19-03416-f010:**
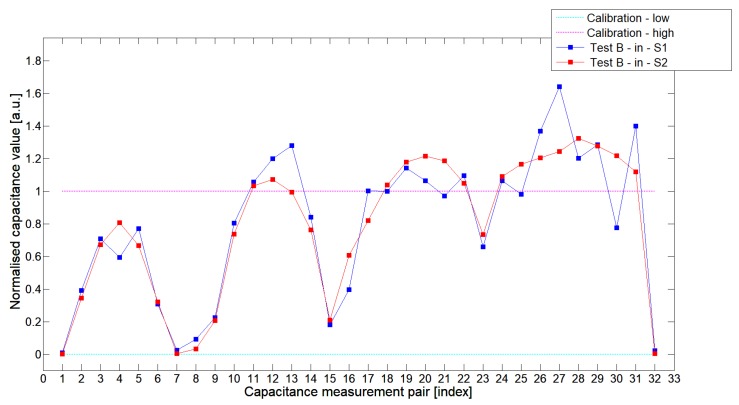
The 1st electrode measurement cycle (S1: 1->32 and S2: 1->32) for Test$B_{in}$ and S1—blue line, S2—red line. Cyan and magenta lines indicate calibration limits (0;1).

**Figure 11 sensors-19-03416-f011:**
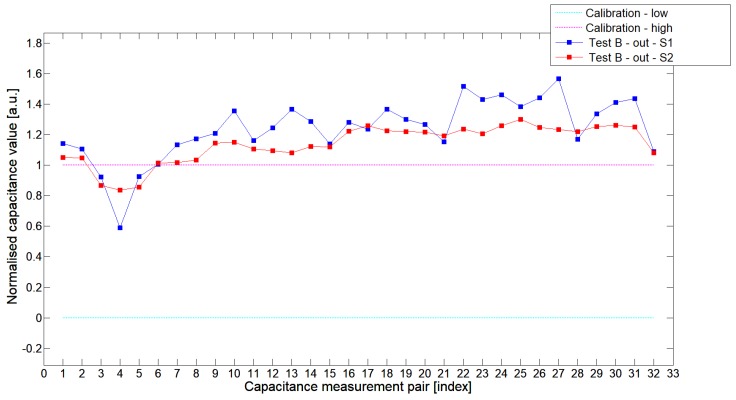
The 1st electrode measurement cycle (S1: 1->32 and S2: 1->32) for Test$B_{out}$ and S1—blue line, S2—red line. Cyan and magenta lines indicate calibration limits (0;1).

**Figure 12 sensors-19-03416-f012:**
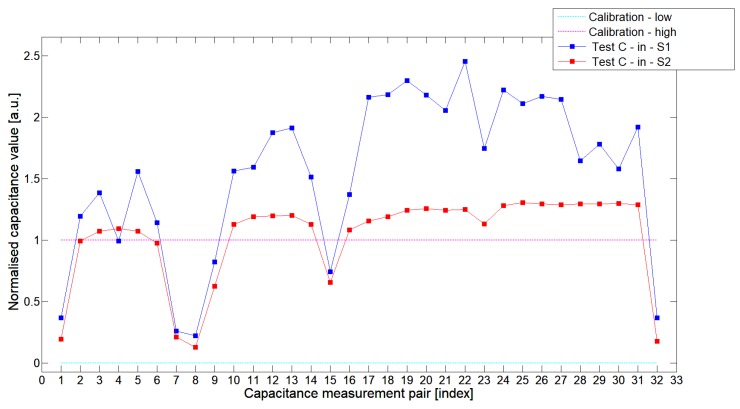
The 1st electrode measurement cycle (S1: 1->32 and S2: 1->32) for Test$C_{in}$ and S1—blue line, S2—red line. Cyan and magenta lines indicate calibration limits (0;1).

**Figure 13 sensors-19-03416-f013:**
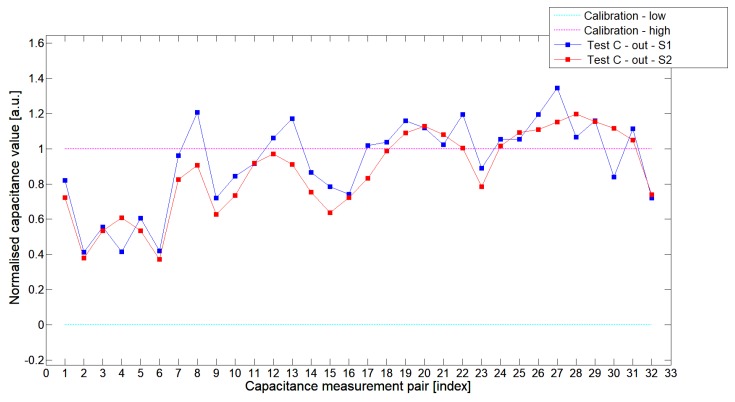
The 1st electrode measurement cycle (S1: 1->32 and S2: 1->32) for Test$C_{out}$ and S1—blue line, S2—red line. Cyan and magenta lines indicate calibration limits (0;1).

**Figure 14 sensors-19-03416-f014:**
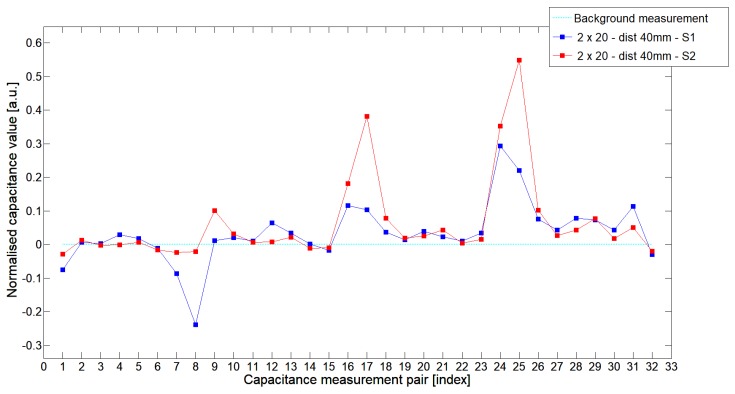
The 1st electrode measurement cycle (S1: 1->32 and S2: 1->32) for Test2x40 and S1—blue line, S2—red line. Cyan line indicates lower calibration limit.

**Figure 15 sensors-19-03416-f015:**
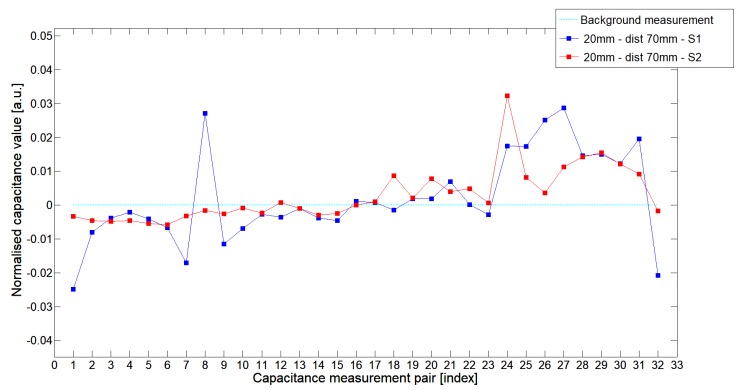
The 1st electrode measurement cycle (S1: 1->32 and S2: 1->32) for Test$20_{P1}$ and S1—blue line, S2—red line. Cyan line indicates lower calibration limit.

**Figure 16 sensors-19-03416-f016:**
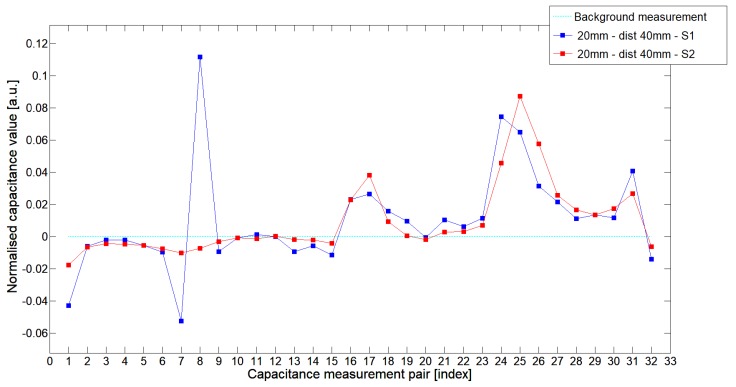
The 1st electrode measurement cycle (S1: 1->32 and S2: 1->32) for Test$20_{P2}$ and S1—blue line, S2—red line. Cyan line indicates lower calibration limit.

**Figure 17 sensors-19-03416-f017:**
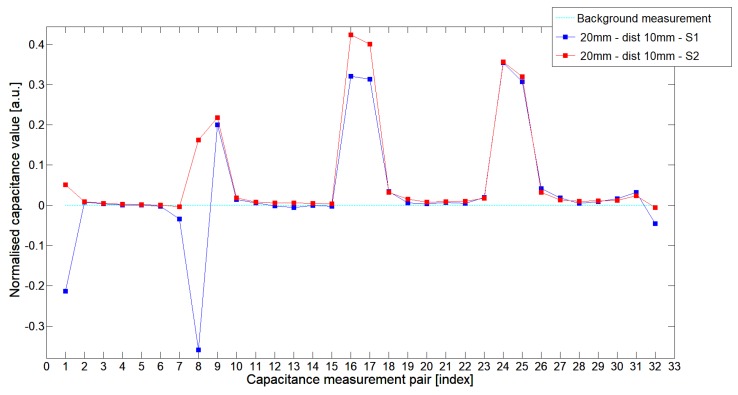
The 1st electrode measurement cycle (S1: 1->32 and S2: 1->32) for Test$20_{P3}$ and S1—blue line, S2—red line. Cyan line indicates lower calibration limit.

**Figure 18 sensors-19-03416-f018:**
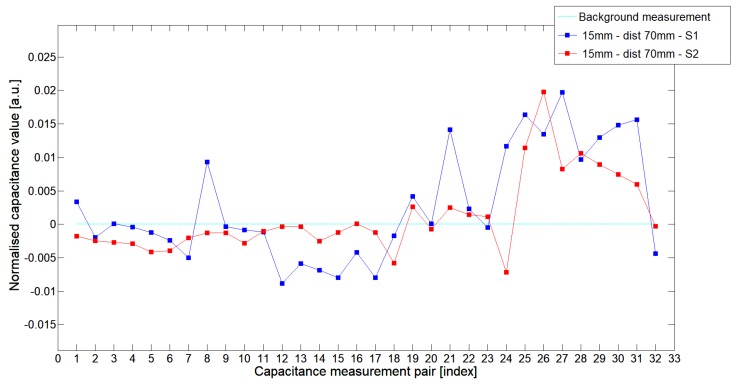
The 1st electrode measurement cycle (S1: 1->32 and S2: 1->32) for Test$15_{P1}$ and S1—blue line, S2—red line. Cyan line indicates lower calibration limit.

**Figure 19 sensors-19-03416-f019:**
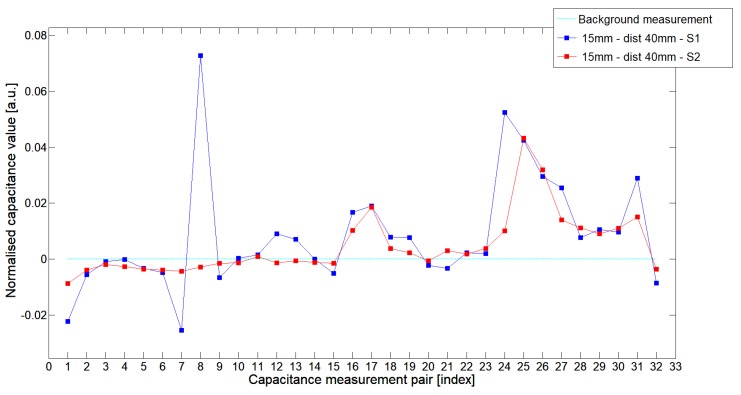
The 1st electrode measurement cycle (S1: 1->32 and S2: 1->32) for Test$15_{P2}$ and S1—blue line, S2—red line. Cyan line indicates lower calibration limit.

**Figure 20 sensors-19-03416-f020:**
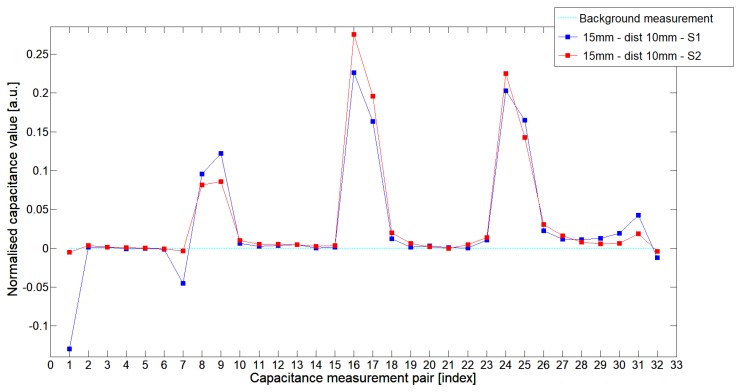
The 1st electrode measurement cycle (S1: 1->32 and S2: 1->32) for Test$15_{P3}$ and S1—blue line, S2—red line. Cyan line indicates lower calibration limit.

**Figure 21 sensors-19-03416-f021:**
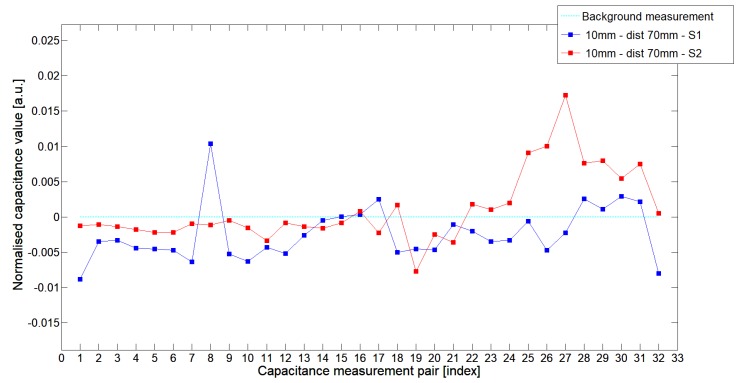
The 1st electrode measurement cycle (S1: 1->32 and S2: 1->32) for Test$10_{P1}$ and S1—blue line, S2—red line. Cyan line indicates lower calibration limit.

**Figure 22 sensors-19-03416-f022:**
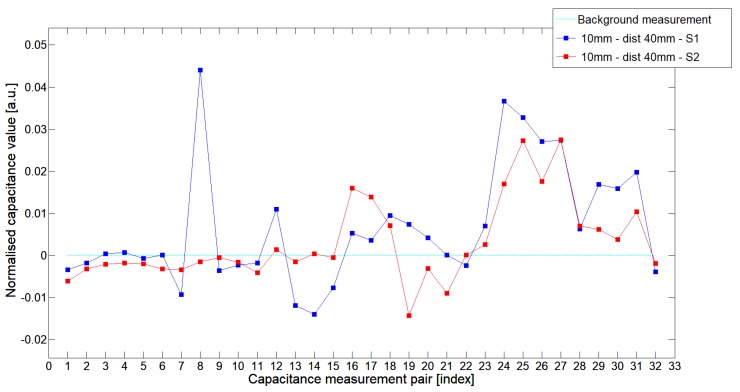
The 1st electrode measurement cycle (S1: 1->32 and S2: 1->32) for Test$10_{P2}$ and S1—blue line, S2—red line. Cyan line indicates lower calibration limit.

**Figure 23 sensors-19-03416-f023:**
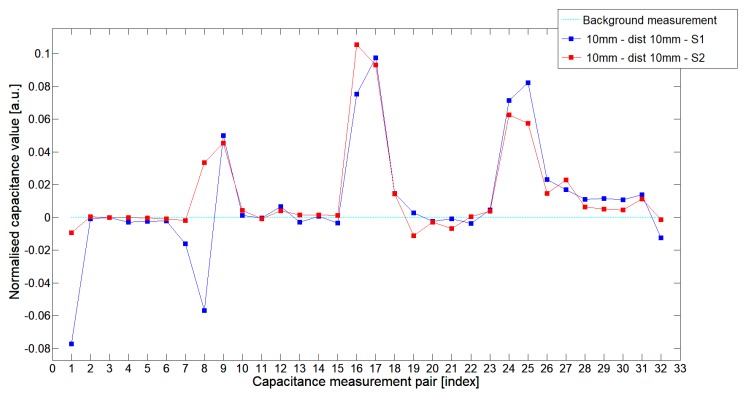
The 1st electrode measurement cycle (S1: 1->32 and S2: 1->32) for Test$10_{P3}$ and S1—blue line, S2—red line. Cyan line indicates lower calibration limit.

**Table 1 sensors-19-03416-t001:** Configuration of PE granules filler low-contrast Test*A*, Test*B*, and Test*C* phantoms.

Config	Test*A*	Test*B*	Test*C*
inside	Phantom Test$A_{in}$	Phantom Test$B_{in}$	Phantom Test$C_{in}$
outside	Phantom Test$A_{out}$	Phantom Test$B_{out}$	Phantom Test$C_{out}$

**Table 2 sensors-19-03416-t002:** Results of capacitance measurement (in pF) for selected electrode pairs and all testing phantoms.

RCD	Test$A_{\mathrm{out}}$	Test$A_{\mathrm{in}}$	Test$B_{\mathrm{out}}$	Test$B_{\mathrm{in}}$	Test$C_{\mathrm{out}}$	Test$C_{\mathrm{in}}$
S1-1-2	7.446	7.291	7.460	7.278	7.408	7.336
S1-1-9	6.604	6.536	6.614	6.531	6.616	6.541
S1-1-17	3.926	3.914	3.928	3.911	3.918	3.930
S1-1-25	3.919	3.917	3.921	3.917	3.917	3.928
S2-1-2	4.909	4.626	4.922	4.617	4.827	4.673
S2-1-9	4.577	4.414	4.591	4.410	4.568	4.427
S2-1-17	3.920	3.912	3.921	3.911	3.913	3.916
S2-1-25	3.922	3.920	3.922	3.920	3.919	3.922

**Table 3 sensors-19-03416-t003:** The standard deviation indicators (std) calculated for the 1st electrode measurement cycle, where electrodes: 2->32 were grounded. Where 10, 15, 20, 40 stands for object diameter and P1, P2, P3 determine object positions along sensor radius, P1 = 70 mm, P2 = 35 mm, P3 = 10 mm respectively.

std	2 × 40	20${}_{P1}$	20${}_{P2}$	20${}_{P3}$	15${}_{P1}$	15${}_{P2}$	15${}_{P3}$	10${}_{P1}$	10${}_{P2}$	10${}_{P3}$
S1	0.0883	0.0132	0.0310	0.1403	0.0083	0.0201	0.0728	0.0038	0.0143	0.0349
S2	0.1296	0.0081	0.0223	0.1272	0.0057	0.0110	0.0717	0.0050	0.0097	0.0284
%	−32%	63%	39%	10%	46%	83%	1.5%	−24%	47%	23%

**Table 4 sensors-19-03416-t004:** The mean value indicators (mean) calculated for the 1st electrode measurement cycle (with 2..32). Where 10, 15, 20, 40 stands for object diameter and P1, P2, P3 determine object positions along sensor radius, P1 = 70 mm, P2 = 35 mm, P3 = 10 mm respectively.

mean	2 × 40	20${}_{P1}$	20${}_{P2}$	20${}_{P3}$	15${}_{P1}$	15${}_{P2}$	15${}_{P3}$	10${}_{P1}$	10${}_{P2}$	10${}_{P3}$
S1	0.0330	0.0020	0.0099	0.0332	0.0011	0.0083	0.0299	−0.0024	0.0031	0.0097
S2	0.0632	0.0028	0.0091	0.0683	0.0027	0.0046	0.0363	0.0011	0.0067	0.0143
%	48%	29%	−8%	51%	45%	−80%	−18%	>100%	>100%	68%

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
