# Peer review of "3D-Printed Multilayer Sensor Structure for Electrical Capacitance Tomography"

_sensors, 2019, doi:10.3390/s19153416_

Round 1
Reviewer 1 Report
This paper describes “3D-printed Multilayer Sensor Structure for Electrical Capacitance Tomography”. However, this manuscript is not satisfied with academic paper standard. This reviewer suggests strongly this manuscript should be improved based on the following points.
At the end of introduction, clear objectives should be written.
Fig. 1 is not necessary.
Many references are listed. But they are not focused on the originality.
3D-“printed” Multilayer Sensor Structure could be original. But, many papers already were published in terms of 3D structure.
Many experimental results are shown. But just showing results are not academic paper. It looks like a technical papers. Academic discussion is not sufficient.
Table should be changed to graph.
Letters in figures are too small, no unit, no symbol.
Reviewer 2 Report
The authors used the 3D-printing technique for ECT sensor development. The topic is interesting. It will be helpful for the researchers in the ECT field. But, some improvements are requried.
Firstly, the advantages of the 3D-printed ECT sensor should clarified in detail. According to the manuscript, the 3D-printing was used for building the mechanical support of ECT sensor to achieve 0.4mm thickness. However, the sub millimeter thickness can also be achieved by coating the ECT electrodes with non-conductivity materials, which is also helpful ECT against the high temperature and high pressure condition. On the other hand, the ECT sensors made by PLA would not survival in high temperature and high pressure condition.
Secondly, the 3D sensor structure is produced by adding consecutive layers of melted substance one by one. However, without the support of the bottom layer, the PLA hot melt deposition will be easy to collapse, especially when the thickness is submillimeter. Please gives the details of overcoming this problem.
Thirdly, whether the thin wall of the tube covered by the electrode will decrease the hardness of the sensor, thus makes the sensor more vulnerable.
Fourthly, in the results and discussion section, please explain why the 3D printed sensors outperform the traditional sensors. In other words, what lead the performance improvement of the printed sensor.
Moreover, there are lots of typo required to be corrected. e.g. Page 1, line 17, 'senor'.
Reviewer 3 Report
This paper proposes a new approach for fabrication of electrical capacitance tomography (ECT) sensor based on the modern efficient 3D printing technique. The main features of the proposed sensing structures are thin walls between the electrode arrays and testing volume as well as original structure of internal vertical and horizontal screening system. The proposed 3D printed 3D ECT sensor has been compared to the conventional analog investigated earlier.
The introduction section of the paper proposes wide review on recent techniques, algorithms and industrial applications of ECT with more than 30 corresponding references. The fundamentals of ECT, the process of 3D printing, the experimental setup and the methodic of measurements and analysis are briefly considered. The experimental results are detail described and presented in Figures justify the advantages of the proposed method.
The reviewer suggests that this paper will be very useful for specialists in the field of electrical capacitance tomography systems. This paper can be accepted after following minor corrections.
1. The order of references in the text is incorrect. The first reference has the number [11], the second one [23], etc.
2. In expression (6), the private derivative is written incorrectly (dj/dn) and must be rewritten as ¶j/¶n.
3. Expression (6) is erroneous. In the left part Ik is declared as the electric current, but the right part of (6) express an electric charge.
4. There is no explanation for Ek after the expression (6).
5. In the third column of Table 1 “Phantom TestBin” must be rewritten as “Phantom TestСin”.
6. In the third column of Table 1 “Phantom TestBout” must be rewritten as “Phantom TestСout”.
7. In the line before the expression (3) a bracket is missing.

Round 2
Reviewer 1 Report
I think it is revised to some extent.
But it is not enough for the publication because papaer quality is not enough.
Just for isntane, x and y axis of a graph needs symbol and unit even though it is nondimentional.
Too smll letters in a graph are not visible.
Figures technical aspects are poor.
Just for instance, fig 7 needs x and y axis.
It is not clear where the possion of rod is, how much layer you put.
and so on.
I hope the author would make a standard quality of academic paper
Author Response
Regarding the issues raised in the second-round review the given paper improvements have been provided:
1. All the graphs were completely rebuilt for better description, legend and unit symbols visibility.
2. Figure 7 description was supplemented by additional explanation of geometrical properties of test objects.